# Improving the Quality of Care for Cancer Patients through Oncological Second Opinions in a Comprehensive Cancer Center: Feasibility of Patient-Initiated Second Opinions through a Health-Insurance Service Point

**DOI:** 10.3390/diagnostics13213300

**Published:** 2023-10-25

**Authors:** Carla E. Schulmeyer, Matthias W. Beckmann, Peter A. Fasching, Lothar Häberle, Henriette Golcher, Frank Kunath, Bernd Wullich, Julius Emons

**Affiliations:** 1Department of Gynecology and Obstetrics, Erlangen University Hospital, Comprehensive Cancer Center Erlangen-EMN, Friedrich Alexander University of Erlangen–Nuremberg, 91054 Erlangen, Germany; 2Biostatistics Unit, Department of Gynecology and Obstetrics, Comprehensive Cancer Center Erlangen-EMN, Friedrich Alexander University of Erlangen–Nuremberg, 91054 Erlangen, Germany; 3Department of Surgery, Erlangen University Hospital, Comprehensive Cancer Center Erlangen-EMN, Friedrich Alexander University of Erlangen–Nuremberg, 91054 Erlangen, Germany; 4Department of Urology, Klinikum Bayreuth GmbH, 95445 Bayreuth, Germany; 5Department of Urology and Pediatric Urology, Erlangen University Hospital, Comprehensive Cancer Center Erlangen-EMN, Friedrich Alexander University of Erlangen–Nuremberg, 91054 Erlangen, Germany; bernd.wullich@uk-erlangen.de

**Keywords:** second medical opinion, gynecologic cancer, urologic cancer, sarcoma, gastrointestinal cancer

## Abstract

Background: To improve the quality and cost-effectiveness of care, cancer patients can obtain a second medical opinion on their treatment. Validation of the diagnostic procedure (e.g., imaging), diagnosis, and treatment recommendation allows oncological therapy to be applied in a more targeted way, optimizing interdisciplinary care. This study describes patients who received second opinions at the Comprehensive Cancer Center for Erlangen–Nuremberg metropolitan area in Germany over a 6-year period, as well as the amount of time spent on second-opinion counseling. Methods: This prospective, descriptive, single-center observational study included 584 male and female cancer patients undergoing gynecological, urologic, or general surgery who sought a second medical opinion. The extent to which the first opinion complied with standard guidelines was assessed solely descriptively. Results: The first opinion was in accordance with the guidelines and complete in 54.5% of the patients, and guideline compliant but incomplete in 13.2%. The median time taken to form a second opinion was 225 min, and the cancer information service was contacted by patients an average of eight times. Conclusions: The initial opinion was guideline compliant and complete in every second case. Without a second opinion, the remaining patients would have been denied a guideline-compliant treatment recommendation. Obtaining a second opinion gives patients an opportunity to receive a guideline-compliant treatment recommendation and enables them to benefit from newer, individualized therapeutic approaches in clinical trials. Establishing patient-initiated second opinions via central contact points appears to be a feasible option for improving guideline compliance.

## 1. Introduction

Numerous national and international guidelines are available for diagnosis, treatment, and follow-up for most cancer entities. Adherence to evidence-based guidelines offers clear benefits to patients, as it is associated with an improved outcome [1]. A guideline-compliant diagnosis for each patient is a prerequisite for being able to offer guideline-compliant therapy. Structurally standardized procedures already start with the assessment of early findings and precursor lesions, such as those in the breast or cervix [2,3].

The provision of oncological therapy in specialized cancer centers has also demonstrated an enhanced outcome for patients with various types of cancer [4,5].

However, the proportion of patients who receive treatment in accordance with oncological guidelines is not known. Not every patient is able to attend a specialized oncological center for various reasons, such as a lack of awareness regarding their availability or the distances required to travel to the specialized center [6].

The opportunity to obtain a second opinion on medical issues is currently becoming increasingly important given the availability of new, individualized treatment options and the patients’ capacities to independently seek information about cancer treatment, particularly through the Internet [7]. The advantages of a second opinion include obtaining information about modern and more experimental treatment options and improving the quality and cost-effectiveness of cancer care.

The benefits of seeking a second opinion and the underlying motivations are subjects of ongoing research. Seeking a second opinion is associated with the avoidance of both overtreatment and undertreatment [8]. Patient satisfaction is reported to be significantly higher in patients who seek a second opinion [9], as it often leads to improved diagnosis and/or treatment [10]. The motivations for seeking a second opinion have also been investigated; in 70% of cases, patients seek more information and/or reassurance after receiving the initial treatment recommendation [11]. There is a strong correlation between disease-related fear or distress and patients seeking a second opinion [12].

The legal requirements for obtaining a second medical opinion in Germany are governed by the Law on Improving Statutory Health Insurance Care (*GKV-Versorgungsstärkungsgesetz*, 2015) [13], and regulations on centers providing such second opinions apply, affecting reimbursement by health insurance companies [13]. Under the statutory health insurance system in Bavaria (*Allgemeine Ortskrankenkasse*, AOK, Munich, Germany), insured individuals can obtain a second medical opinion for oncological diseases via an interdisciplinary tumor board. The pilot project investigated in the present study was initiated pursuant to the German Social Security Code, Book V, Section 63, paragraph 1 [14].

Research has shown that cancer patients experience a better outcome when treated in accordance with national guidelines and national treatment recommendations based on the latest developments in clinical trials [15]. The aim of this prospective study of cancer patients seeking a second opinion was therefore to analyze whether the first opinion that they had received was in accordance with national guidelines.

## 2. Materials and Methods

Eligible patients included men and women who used the AOK (statutory health insurance) service number to contact the cancer information service at Erlangen University Hospital and received a second opinion regarding their cancer therapy. Patients were only included if they had never been admitted to Erlangen University Hospital for this specific type of tumor and had had no outpatient contact with the hospital within the last 3 months. Cancer patients in the fields of urology, gynecology, gastroenterology, and sarcomas were included. This is due to the high caseload in these areas and the presence of specific, validated “Level 3” (evidence-based and consensus-based) national guidelines [16]. Exclusion criteria encompassed patients who contacted the service number for reasons unrelated to seeking a second oncological opinion, patients who did not provide their full cancer history, and patients with malignancies other than in the fields of urology, gynecology, gastroenterology, or sarcoma.

Patient recruitment took place between 1 June 2014, and 31 May 2020. A total of 584 patients met the inclusion criteria and were included in the final analysis (Figure 1).

### 2.1. Study Design

The study procedure was conducted in the following manner: Initially, all patients who contacted the cancer information service via the AOK service number were documented. The patients’ characteristics and their clinical reports were collected during the initial contact—e.g., gender and age as well as the cancer diagnosis. A complete list is provided in Appendix A. The method of counseling offered (by phone, in writing, etc.) and the duration of contact were also documented.

Upon obtaining written consent from the patients, their cases were presented to the interdisciplinary tumor board. The tumor board’s recommendation was then documented on a second opinion form, and the extent of guideline compliance with the first opinion treatment recommendation was assessed. A form stating the interdisciplinary tumor board’s recommendation was sent to the patients. If any questions or uncertainties arose thereafter, patients were able to contact the Comprehensive Cancer Center once again.

After 3 months, follow-up inquiries were made concerning the patient’s current treatment status. If this information was not assessable, the reason for this was identified and documented if possible—e.g., because the patient had died or had withdrawn consent. Based on the information obtained, the extent of the patient’s adherence to the recommendations offered in the second opinion was assessed (this will be reported on elsewhere).

### 2.2. Data Acquisition

*Patients’ approaches to second opinions.* All patients who used the AOK (statutory health insurance) hotline to contact the cancer information service affiliated with the Comprehensive Cancer Center (CCC) for the Erlangen–Nuremberg metropolitan area were documented. During the initial contact, patient characteristics were recorded, including the patient’s gender, age, cancer diagnosis, and tumor stage, at the time of the inquiry. The complete data file is attached in Appendix A. The duration of the phone call was documented. Subsequently, CCC staff sent the patients a consent form and a questionnaire for the second opinion, and the patient was registered with the relevant department’s interdisciplinary tumor board.

*Presentation at the interdisciplinary tumor board.* The patient’s case was presented in the relevant interdisciplinary tumor conference, which includes various disciplines—e.g., gynecology, urology, gastroenterology, pathology, radiology, general surgery, medical oncology, and radiotherapy. Treatment recommendations were discussed on an interdisciplinary basis and were aligned with the standards of the “Level 3” (evidence-based and consensus-based) national guidelines—e.g., for breast, cervical, or vulvar cancer, colorectal cancer, or prostate cancer [16]. The tumor board’s treatment decision and the consistency of the initial recommendation with the national guidelines were documented. Subsequently, all patients received a letter stating the tumor board’s decision. If there were any remaining questions or uncertainties, the patients were able to contact the CCC once again.

*Assessing the degree of agreement between the first and second opinions.* The extent of agreement between the first and second opinions was exclusively assessed for patients in the gynecology, urology, and surgery departments included in the study. This evaluation of the agreement was undertaken by two medical experts. In the event of discrepancies between the two assessors, the decision of the more experienced assessor was used on the basis of his or her clinical experience.

### 2.3. Ethics

This study was approved by the Ethics Committee at Erlangen University Hospital (no. 175_13B) and was conducted in accordance with the ethical principles set out in the Declaration of Helsinki.

### 2.4. Statistical Analysis

Guideline compliance and time spent on second opinion counseling were evaluated descriptively, with frequencies and percentages. Statistical tests were not performed since no group comparisons or interventions were performed. All statistical analyses were performed using the R statistical software package (version 3.6.1, 2019; R Core Team, Vienna, Austria).

## 3. Results

### 3.1. Patient Group

A total of 2342 patients contacted the AOK (statutory health insurance) service number. Among these, 1500 patients received information about a second opinion; the remaining calls were made for various reasons—e.g., a change of address or loss of the patient’s insurance card. Ultimately, 705 patients received a second opinion. The remaining patients did not receive a second opinion—for example, because they did not submit any documents. Among the 705 patients who received a second opinion, 586 had cases relevant for the tumor conferences on gynecological, urological, or gastroenterological cancers or sarcoma that were participating in the analysis. Two patients were excluded due to missing information in the main variables. Figure 1 provides a comprehensive breakdown of the patient recruitment process.

### 3.2. Patient Characteristics

The mean age of the 584 patients with malignancies in the fields of gynecology, urology, gastroenterology, or sarcoma was 61 years (standard deviation of 12.3 years). Among these, 59% of the patients were women (*n* = 343), and 41% were men (*n* = 238).

The primary diagnosis showed no distant metastases in 32.9% of cases (*n* = 188); 17% of the patients (*n* = 97) had distant metastases. The patients were free of tumors in 18.6% of cases (*n* = 106). At the time of consultation, most patients had not yet received any treatment, but treatment was planned in 76.1% (*n* = 392). Primary therapy was being administered to 17% (*n* = 99), and 13.4% (*n* = 78) of the patients were undergoing palliative therapy (Table 1).

The characteristics of all the patients in all departments (*n* = 705) are shown in Table 2.

### 3.3. Guideline Compliance of the First Opinion

For a total of 584 patients seeking a second opinion in the fields of gynecology, urology, or general surgery, the initial opinion was found to be guideline compliant and complete in 54.5% of cases (*n* = 318). The initial opinion was guideline compliant but incomplete in 13.2% (*n* = 77). Guideline compliance was not assessable—e.g., due to missing data—in 12.8% (*n* = 75) of cases, and guidelines were not available for less common tumor entities in 11.5% of cases. The first opinion was found not to be in accordance with the guidelines in 6.8% of cases (*n* = 40). There was no information regarding the tumor entity, or the tumor entity was not part of the evaluation in 1.2% of cases (*n* = 7) (Table 3).

### 3.4. Time Required until a Second Opinion Is Formed

Figure 2 shows the distribution of the total time needed for second opinion counseling. The mean total time needed was 239.8 min (standard deviation: 60.4 min), and the median was 225 min (interquartile range: 200–260 min). In the fastest case, a second opinion took 140 min. In the longest case, 780 min were needed for second-opinion counseling (Table 4). Registration for the tumor board, which was performed by the cancer information staff, took 120 min in each case.

Each patient contact (e.g., in the form of a telephone call or also a written contact) was evaluated as an action, and registration for the interdisciplinary tumor board or the receipt or dispatch of documents was also evaluated as an action.

## 4. Discussion

The aim of this prospective study of cancer patients seeking a second opinion was to analyze whether the first opinion they had received was in accordance with national guidelines. The initial opinion was found to be guideline-compliant and complete in every second case. Without a second opinion, a guideline-compliant treatment recommendation would have been denied to the remaining patients.

Second medical opinions can have a substantial influence on cancer patients’ well-being, treatment, and prognosis. The present study was initiated in order to examine the quality and effectiveness of care for oncological patients during the process of obtaining a second medical opinion through the AOK second opinion project. The report includes the largest group of cancer patients in Germany investigated to date concerning second opinions in Germany.

In the past, observations on second opinions have often focused on interobserver variability of diagnostic procedures, such as histological processing with molecular markers, pathological risk factors, or imaging [17,18,19,20,21]. In today’s clinical routine, most oncological therapy recommendations are generated in interdisciplinary tumor conferences, renowned for their enhancement of patient care and treatment planning [22,23,24]. Investigating therapy recommendations is thus an important aspect of health economics and health-care research.

Few data are available on patient-initiated second medical opinions. One of the greatest obstacles that patients face when seeking a second opinion is managing the doctor–patient relationship [25,26]. A novel approach used in the present trial was that patients were able to request a second opinion through the mediation of the health insurance company, independently of traditional face-to-face consultations. This approach may lower the barrier to seeking a second opinion and does not involve any personal costs for patients, such as transportation to the hospital. In addition, having an objective, neutral point of contact accessible by phone makes the second opinion easily available and can break down psychological barriers for both patients and oncologists.

From the perspective of insurance companies, second opinions serve as valuable tools for providing objective quality assurance. As this study shows, second opinions are more likely to lead to guideline-compliant recommendations, improving the quality and cost-effectiveness of the treatment [2].

In Germany, the certification procedures for oncological cancer centers require clinical trials to be available for patients. Patients are therefore more likely to be able to participate in clinical trials in certified oncological cancer centers [27]. This access affords them the opportunity to receive individualized treatments, including targeted and/or immune therapies, which can contribute to long-term advancements in medical care.

The results of this study are consistent with our group’s previously published data on patient-initiated second opinions [28]. However, it is important to note that the data in the present study encompassed a broader spectrum of oncology patients and included individuals from various oncological centers, yielding similar outcomes.

A previous study reported a higher proportion of first opinions that were in accordance with national guidelines, at 72.2% [29]. Potential reasons for the discrepancy may include different groups of patients since only those with breast cancer were included. Cancer guidelines and certified cancer centers were first established in the field of gynecology for breast cancer. Also, breast cancer therapy is more standardized than for rarer gynecological and other cancers.

Without a second opinion, one-third of the patients would not have had access to fully guideline-compliant treatment recommendations, potentially leading to poorer outcomes, higher costs, and less patient satisfaction. This highlights the need for second opinions issued by qualified interdisciplinary tumor boards, and it also justifies the necessary costs for staffing and infrastructure involved in providing a second opinion. The availability of guideline-compliant treatment could potentially improve an easy-to-access method for initiating and obtaining a second opinion. Examples of accessibility with no personal presentation at the hospital could be by phone or writing, as reported in this study, or an app-based tool. This approach could effectively reduce barriers for patients seeking second opinions, as it does not incur any costs and requires only a minimal time commitment from the patients.

Regarding the patient group, the gender distribution was uneven; 59% of the patients were women. This suggests that women give greater attention to medical concerns and raises the possibility of a potential disadvantage for male patients in patient-initiated healthcare approaches.

An analysis of the patients’ different therapy situations showed that individuals seek second opinions across all stages of treatment. The timing of these consultations was mostly during diagnosis and prior to the initiation of therapy, but also during treatment. The duration and timing of seeking a second opinion are therefore critical factors to consider, as they can impact the commencement of therapy, leading to a potential increase in mortality [30].

The strong demand for second opinions was observed in patients with previous treatment and/or metastases. This underlines the need for second opinions in advanced therapy lines when clinical guidelines offer multiple (chemo)therapeutic options, and suitable therapy recommendations are especially crucial for individual patients. The demand for second opinions was also substantial both before and during new treatments, either to confirm the current therapy or to have validated therapy recommendations for the subsequent treatment.

The sometimes large expenditure of time required until a second opinion is obtained is justified in any case if it results in a guideline-compliant therapy recommendation for the patient.

### Strengths and Limitations

This study includes the largest cohort of patients seeking a second opinion that has been investigated to date in Germany, with different tumor entities from different oncological centers (urology, gastroenterology, gynecology, and sarcoma).

The limitations of the analysis include the degree of comparability of this patient population with those described in other studies. All included patients were in the same insurance class, well informed about the option of requesting a second opinion through the insurance company, and mostly in adequate health. The second opinion in this study was patient-initiated and therefore do not include certain cohorts, such as patients with poorer health status. For the latter group, it is known that patients who rate their health status as severe are less likely to participate in research surveys [31]. However, for such patients with considerable periods of previous treatment, therapy recommendations are challenging, and a second opinion could be especially beneficial.

The patterns and outcomes of second opinions may vary in different patient populations. For example, patients with private health insurance, those who are less well informed, and those in poorer health may offer different results. Nonetheless, this study offers novel and clinically relevant information on cancer patients seeking a second opinion as well as on decision-making in cancer treatment.

Furthermore, another limitation of this study lies in the absence of a more detailed description of pathological and clinical features. While our research has examined and elucidated various aspects of second opinions in cancer patients, this omission is a result of a deliberate focus on broader patterns and mechanisms. The findings presented herein are limited in their scope and do not provide insights into the pathological underpinnings that may be relevant in certain contexts.

## 5. Conclusions

Every oncological patient has the right to seek a second medical opinion, which often provides added value for the patient’s treatments. Obtaining a second opinion offers patients the opportunity to receive a guideline-compliant therapy recommendation and to benefit from new treatment approaches within the framework of clinical studies. It can, however, be a time-consuming process for both patients and oncological staff.

This study presents research on patient-initiated second opinions and includes a large group of oncological patients with different malignancies, thanks to cooperation between three oncological centers in the fields of urology, general surgery, and gynecology. The easy-to-access method for obtaining a second opinion reported in this study could potentially improve the availability of guideline-compliant therapy.

Healthcare providers should not discourage cancer patients from exploring the option of seeking a second opinion, particularly in cases involving complex therapy decisions, prior treatment history, or metastases. These results emphasize the need for accessible methods for obtaining second opinions, which can contribute to the availability of guideline-compliant treatments.

Future research endeavors should focus on assessing patients’ adherence to second opinions and understanding the factors influencing their decisions to seek or not seek such consultations.

## Figures and Tables

**Figure 1 diagnostics-13-03300-f001:**
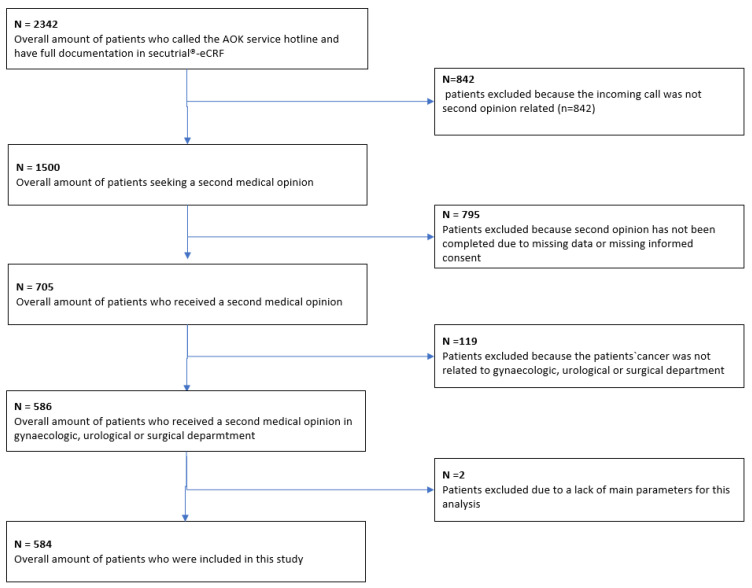
Flow chart for patient selection.

**Figure 2 diagnostics-13-03300-f002:**
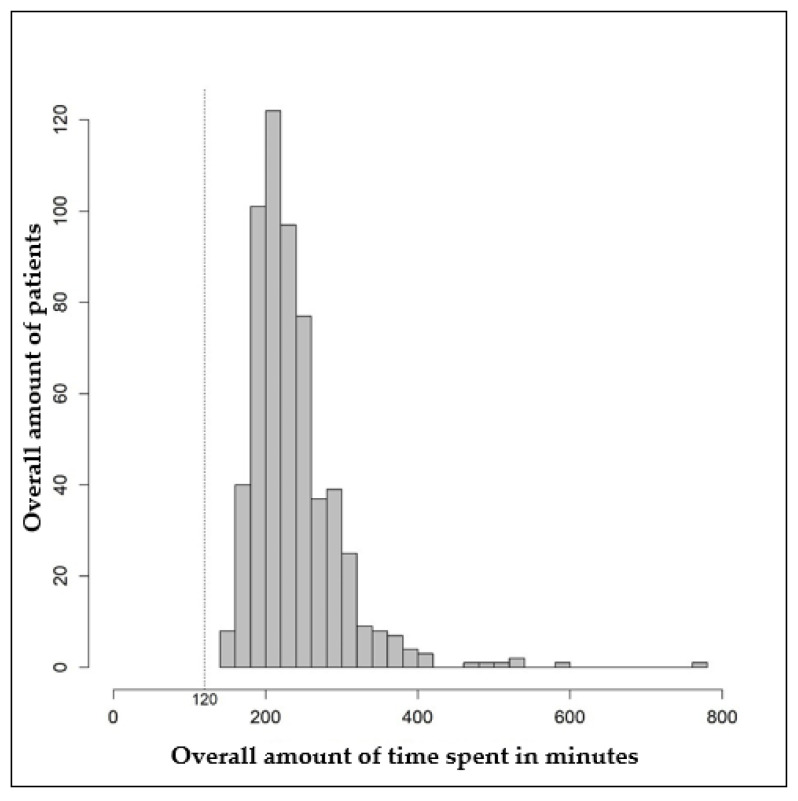
Histogram of the total time in minutes required for second opinion counseling.

**Table 1 diagnostics-13-03300-t001:** Patient and tumor characteristics at first contact with cancer information center for patients in the clinic of surgery, urology or gynaecology (*n* = 584). The values represent mean (MW) with standard deviation (SD), median with interquartile range or frequency with percent.

Characteristics	Expression	
Age	MW (SD) in years	61.0 (12.3)
	Median (interquartile range), in years	61 (53–70)
	*Missing values*	0
Gender	Female	343 (59.0)
	Male	238 (41.0)
	*Missing values*	*3*
Tumor status/situation	Diagnosis	0 (0.0)
	Tumor-free	106 (18.6)
	Suspicion of tumor	6 (1.1)
	Primary tumor	188 (32.9)
	Primary tumor/distant metastases	97 (17.0)
	Distant metastases	137 (24.0)
	suspected recurrence	14 (2.5)
	Recurrence	23 (4.0)
	Unknown	0 (0.0)
	Deceased	0 (0.0)
	*Missing values*	*13*
Therapy status	No therapy/therapy planned	392 (67.1)
	Under primary therapy	99 (17.0)
	After primary therapy	15 (2.6)
	Palliative therapy	78 (13.4)
	No therapy, since deceased	0 (0.0)
	Unknown	0 (0.0)
	*Missing values*	0

**Table 2 diagnostics-13-03300-t002:** Patient and tumor characteristics at first contact with cancer information for patients at all clinics (*n* = 705). The values represent mean (MW) with standard deviation (SD), median with interquartile range or frequency with percent.

Characteristic	Expression	
Age	MW (SD), in years	61.0 (12.5)
	Median (interquartile range), in years	62 (53–70)
	*Missing values*	*1*
Gender	Female	404 (57.6)
	Male	297 (42.4)
	*Missing values*	*4*
Tumor status/situation	First Diagnosis	0 (0.0)
	Tumor-free	121 (17.6)
	Suspicion of (V.a.) tumor	7 (1.0)
	Primary tumor	242 (35.2)
	Primary tumor/distant metastases	120 (17.5)
	Distant metastases	147 (21.4)
	suspected recurrence	17 (2.5)
	Recurrence	33 (4.8)
	Unknown	0 (0.0)
	Deceased	0 (0.0)
	*Missing values*	*18*
Therapy status	No therapy/therapy planned	482 (68.5)
	Under primary therapy	117 (16.6)
	After primary therapy	16 (2.3)
	Palliative therapy	89 (12.6)
	No therapy, since deceased	0 (0.0)
	Unknown	0 (0.0)
	*Missing values*	*1*

**Table 3 diagnostics-13-03300-t003:** Guideline accordance of the first opinion (*n* = 584).

Guideline Accordance of the First Opinion	*n*	%
First opinion is complete and in accordance with guidelines	318	54.5
First opinion is guideline-compliant, but incomplete	77	13.2
First opinion is not in line with guidelines	40	6.8
First opinion is not assessable/missing data	75	12.8
No guideline available for tumor entity	67	11.5
no information/tumor entity not part of the evaluation	7	1.2

**Table 4 diagnostics-13-03300-t004:** Duration of contacts or actions per patient, broken down by type of contact. Data shown in minutes (*n* = 584). Every patient contact (e.g., in the form of a telephone call or also a written contact) was evaluated as an action, but also the registration for the interdisciplinary tumour board or the receipt or dispatch of documents was evaluated as an action.

Contact Type	Minimum	25% Percentile	Median	Mean Value	75% Percentile	Maximum
Individual letter/e-mail	0	10	20	17.9	20	140
Sending information by letter	0	0	5	4.1	5	15
Sending information by mail/fax	0	0	5	4.8	5	30
Return of documents	0	0	10	7.6	10	40
Tumour board registration	0	120	120	124.1	120	240
No counselling/contact	0	0	5	3.9	10	15
Check tumour board result	0	5	5	5.0	5	10
Phone call	0	45	60	71.7	90	410
**Total time spent**	**140**	**200**	**225**	**239.8**	**260**	**780**

## Data Availability

The datasets are available from the corresponding authors upon reasonable request.

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
