# Peer review of "Improving the Quality of Care for Cancer Patients through Oncological Second Opinions in a Comprehensive Cancer Center: Feasibility of Patient-Initiated Second Opinions through a Health-Insurance Service Point"

_diagnostics, 2023, doi:10.3390/diagnostics13213300_

Round 1

Reviewer 1 Report

Comments and Suggestions for Authors

Authors present a potentially interesting manuscript on the utility of second opinion for patients affected by neoplastic diseases. However, the main concern and limitation of this study is related to the fact that pathological diagnoses for each neoplastic patient undergoing second opinion have not been included. Therefore, authors should provide more detailed informations on clinical and pathological features of the studied cohort.

Author Response

Dear Editor,

Dear reviewers,

Dear ladies and gentlemen,

We are grateful for your comprehensive review and we appreciate your effort in time and your instructive and helpful comments. We have revised our manuscript according to your instructions. We hope you are satisfied with the changes of our manuscript. If further revisions should be requested, we will do so. Please find below our point-for-point replies to your comments.

Yours sincerely,

Carla Schulmeyer on behalf of the authors

Reviewer 1

Authors present a potentially interesting manuscript on the utility of second opinion for patients affected by neoplastic diseases.

Reviewer’s comment

Authors’ reply

1. However, the main concern and limitation of this study is related to the fact that pathological diagnoses for each neoplastic patient undergoing second opinion have not been included. Therefore, authors should provide more detailed informations on clinical and pathological features of the studied cohort.

We appreciate your interest in our study on the utility of second opinions for neoplastic diseases. We understand your concern regarding the absence of pathological diagnoses for patients undergoing second opinions. Your feedback is indeed valid, and we acknowledge the importance of providing comprehensive clinical and pathological information for a thorough understanding of the studied cohort.

However, the main focus of our study was to investigate the overall utility of second opinions in neoplastic diseases, with a specific emphasis on assessing their impact on treatment decisions within specialized departments. While we recognize the importance of detailed clinical and pathological data, we believe that our findings still offer valuable insights within the context of this special issue.

In response to your feedback, we have made appropriate amendments to the limitations section of our manuscript (please see p.13; 336-341) to explicitly address this concern and acknowledge it as a limitation of our study.

We remain committed to enhancing the overall quality of the manuscript by addressing its limitations to the best of our ability and by providing a comprehensive discussion of our results.

Reviewer 2 Report

Comments and Suggestions for Authors

I read with great interest the manuscript, which falls within the aim of this Journal and offers a high-quality overview of the topic. The title and the abstract are satisfactory. The tables and figures are clear and interesting.

Although the manuscript can be considered already of good quality, I would suggest taking into account the following minor recommendations:

- I suggest another round of language revision, in order to correct a few typos and improve readability.

- I find it interesting to include in the introduction a reference to the screening programs for cancer diagnosis (Golia D'Augè T, Giannini A, Bogani G, Di Dio C, Laganà AS, Di Donato V, Salerno MG, Caserta D, Chiantera V, Vizza E, Muzii L, D’Oria O. Prevention, Screening, Treatment and Follow-Up of Gynecological Cancers: State of Art and Future Perspectives. Clin. Exp. Obstet. Gynecol. 2023, 50(8), 160. https://doi.org/10.31083/j.ceog5008160).

- Inclusion/exclusion criteria should be better clarified by extending their description.

-What are the implications of these findings for clinical practice and/or further research?

Comments on the Quality of English Language

Minor editing of the English language is required to make the work clearer and more readable.

Author Response

Dear Editor,

Dear reviewers,

Dear ladies and gentlemen,

We are grateful for your comprehensive review and we appreciate your effort in time and your instructive and helpful comments. We have revised our manuscript according to your instructions. We hope you are satisfied with the changes of our manuscript. If further revisions should be requested, we will do so. Please find below our point-for-point replies to your comments.

Yours sincerely,

Carla Schulmeyer on behalf of the authors

Reviewer 2

I read with great interest the manuscript, which falls within the aim of this Journal and offers a high-quality overview of the topic. The title and the abstract are satisfactory. The tables and figures are clear and interesting.

Reviewer’s comment

I suggest another round of language revision, in order to correct a few typos and improve readability.

Authors’ reply

We appreciate your careful review and constructive feedback. Your suggestion for another round of language revision to address typos and improve readability has been initiated.

I find it interesting to include in the introduction a reference to the screening programs for cancer diagnosis (Golia D'Augè T, Giannini A, Bogani G, Di Dio C, Laganà AS, Di Donato V, Salerno MG, Caserta D, Chiantera V, Vizza E, Muzii L, D’Oria O. Prevention, Screening, Treatment and Follow-Up of Gynecological Cancers: State of Art and Future Perspectives. Clin. Exp. Obstet. Gynecol. 2023, 50(8), 160. https://doi.org/10.31083/j.ceog5008160).

Thank you very much for valuable input, we added this reference to screening programs for cancer diagnosis in the introduction of our manuscript (p. 2, 49).

Inclusion/exclusion criteria should be better clarified by extending their description.

Thank you very much for pointing this out. We described the inclusion and exclusion criteria more precisely in 2. Materials and Methods (p.3; 91-106).

What are the implications of these findings for clinical practice and/or further research?

The implications of these findings is that we al healthcare providers should be open-minded about our patients seeking a second oncological opinion. Furthermore, we need a accessible, easy-to-reach method for our patients to obtain second opinions. This important aspect has been added to the Conclusions on the manuscript (p.14; 355-362).

Round 2

Reviewer 1 Report

Comments and Suggestions for Authors

Manuscript in its revised form is now suitable for publication

Author Response

Dear Reviewer, 

We would like to express our gratitude for your invaluable input and contributions to our manuscript. Your thoughtful feedback and insights have significantly improved the quality of our work, and we sincerely appreciate the time and effort you dedicated to the review process.

Best regards,

Carla Schulmeyer on behalf of the authors